# Learning Multi-agent Behaviors from Distributed and Streaming Demonstrations

**Shicheng Liu & Minghui Zhu**
School of Electrical Engineering and Computer Science
Pennsylvania State University
University Park, PA 16802, USA
{sf15539,muz16}@psu.edu

## Abstract

This paper considers the problem of inferring the behaviors of multiple interacting experts by estimating their reward functions and constraints where the distributed demonstrated trajectories are sequentially revealed to a group of learners. We formulate the problem as a distributed online bi-level optimization problem where the outer-level problem is to estimate the reward functions and the inner-level problem is to learn the constraints and corresponding policies. We propose a novel "multi-agent behavior inference from distributed and streaming demonstrations" (MA-BIRDS) algorithm that allows the learners to solve the outer-level and inner-level problems in a single loop through intermittent communications. We formally guarantee that the distributed learners achieve consensus on reward functions, constraints, and policies, the average local regret (over $N$ online iterations) decreases at the rate of $O(1/N^{1-\eta_1} + 1/N^{1-\eta_2} + 1/N)$, and the cumulative constraint violation increases sub-linearly at the rate of $O(N^{\eta_2} + 1)$ where $\eta_1, \eta_2 \in (1/2, 1)$.

## 1 Introduction

Multi-agent systems (MASs) are an effective tool to model networked systems where multiple entities interact with each other to reach certain goals. Due to the lack of a centralized authority, the data in MASs is usually distributed [1, 2]. Therefore, distributed learning is desired for MASs where machine learning models are trained over distributed data sets. Current works on distributed learning include distributed supervised learning [3, 4], distributed unsupervised learning [5, 6], distributed reinforcement learning [7, 8], etc. Recently, distributed learning is applied to learn the behaviors in MASs from distributed experts' demonstrations via inverse reinforcement learning (IRL) [9].

In IRL [10, 11, 12, 13, 14], a learner aims to learn a policy that imitates the expert behaviors in the demonstrations by first learning a reward function. Multi-agent IRL [15, 16, 17] extends IRL to MASs where the reward functions and policies of a group of experts are learned. However, current works on multi-agent IRL have the following two limitations: (i) As mentioned, the demonstration data in MASs is usually distributed while these works assume that there is a centralized learner which can obtain all the demonstrations. (ii) There are usually many underlying constraints in MASs, e.g., avoiding collision with each other and obstacles, and thus learning a reward function combined with a set of constraints is better than learning a single reward function in terms of explaining the experts' behaviors [18, 19]. Therefore, distributed inverse constrained reinforcement learning (D-ICRL) [9] is proposed where a group of learners cooperatively learn the behaviors in an MAS by estimating the experts' reward functions and constraints and each learner can only access a local demonstration set.

While D-ICRL performs over pre-collected distributed data, recent applications [20, 21] of IRL motivate the need for algorithms that can learn from sequentially revealed demonstrations and continuously improve the learned models. For example, inferring a person's intent by observing her

ongoing daily routine [20] and updating evasion strategies by continuously observing the patrollers' behaviors [21]. However, D-ICRL is not efficient to deal with streaming data because it has a double-loop learning structure where the outer loop is to update the constraints and the inner loop needs to find a corresponding reward estimate. While the computation overhead of this double-loop structure is reluctantly acceptable in offline settings, it is too time-consuming for streaming data since the computation may not be finished before the next data arrives. As D-ICRL is the most important baseline of our work, we include a section in Appendix to summarize our improvements from it.

**Related works and our improvements**. Current theoretical works on online IRL [20, 21] only consider linear reward functions and thus cast the problem as online convex optimization. While these approaches achieve sub-linear regret, their analysis does not hold for non-linear reward functions (e.g., neural networks) as the corresponding objective functions can be non-convex. To quantify the algorithmic performance on online non-convex optimization, "local regret" [22, 23] is proposed which, at each online iteration, quantifies the gradient norms of the average sum of some already-revealed loss functions. Current state-of-the-arts [22, 23] use follow-the-leader-based methods to minimize the local regret. While their methods can achieve the tight regret bound, the follow-the-leader-based methods require multiple gradient descent steps until reaching a near-stationary point at each online iteration. However, if the data arrival rate is fast, the computation may not be done before the next data arrives. To alleviate the computation burden at each online iteration, we use an online gradient descent (OGD) method which only updates the decision variable by one gradient descent step at each online iteration. To the best of our knowledge, no works can quantify the local regret of OGD.

Inspired by [9, 24], we formulate a distributed online bi-level optimization problem where the learners cooperatively learn the reward functions in the outer-level problem and the constraints and corresponding policies in the inner-level problem. Papers [9, 24, 25] use double-loop methods [26, 27, 28] to solve their bi-level optimization problems where they first find a (sub)optimal solution of the inner-level problem in a faster loop and then solve the outer-level problem in a slower loop. However, when it comes to streaming data, the double-loop method needs multiple steps for the inner-level problem before updating the decision variable of the outer-level problem, which can be too slow to finish the update of the outer decision variable when the data is revealed in a fast speed. Therefore, we use a single-loop method which updates both the outer and inner decision variable only once at each iteration. Notice that the state-of-the-arts on single-loop bi-level optimization [29, 30] cannot be directly applied to our problem because they are centralized and require the inner objective function to be strongly convex while our problem does not have these properties.

**Contribution statement**. Our contributions are threefold. First, we consider the problem where a group of learners cooperatively recover the policies by estimating the reward functions and constraints from distributed streaming demonstrations of cooperative experts. We formulate this "multi-agent behavior inference from distributed and streaming demonstrations" (MA-BIRDS) problem as a distributed online bi-level optimization problem. Second, we propose a novel distributed online gradient descent algorithm for the learners to learn the reward functions, constraints, and the corresponding policies in a single loop where the decision variables of both the outer-level and inner-level problems are updated only once at each online iteration. Third, we prove that the distributed learners achieve consensus in reward functions, constraints, and policies, respectively, at the rate of $O(1/N^{\eta_1} + \bar{\epsilon}^N)$, $O(1/N^{\eta_2} + \bar{\epsilon}^N)$, and $O(1/N^{\eta_1} + 1/N^{\eta_2} + \bar{\epsilon}^N)$ where $\bar{\epsilon} \in (0, 1)$ and $\eta_1, \eta_2 \in (1/2, 1)$. The local regret averaged over $N$ iterations decreases at the rate of $O(1/N^{1-\eta_1} + 1/N^{1-\eta_2} + 1/N)$ and the cumulative constraint violation grows sub-linearly at the rate of $O(N^{\eta_2} + 1)$. Moreover, if the reward functions are linear, we prove that the average cumulative reward difference between the experts and learners diminishes at the rate of $O(1/N^{1-\eta_1} + 1/N)$.

## 2 Model

This section presents the models of the experts and learners.

**Experts**. There are $N_E$ experts whose decision making is modeled as a constrained Markov game (CMG) [31]. A CMG $(\mathcal{S}, \mathcal{A}, \gamma, T, P_0, P, r_E, c_E, b)$ consists of a state set $\mathcal{S} \triangleq \prod_{i=1}^{N_E} \mathcal{S}^{(i)}$, an action set $\mathcal{A} \triangleq \prod_{i=1}^{N_E} \mathcal{A}^{(i)}$, a discount factor $\gamma$, a time horizon $T$, and an initial state distribution $P_0$. The state transition function is $P$ and $P(s'|s, a)$ represents the probability of transitioning to state $s'$ from $s$ by taking action $a \triangleq (a^{(1)}, \cdots, a^{(N_E)})$. Expert $i$'s reward function is $r_E^{(i)} : \mathcal{S} \times \mathcal{A} \to \mathbb{R}$ and the

experts are cooperative, i.e., $r_E \triangleq \sum_{i=1}^{N_E} r_E^{(i)}$. The cost function of expert $i$ is $c_E^{(i)} \triangleq (\omega_E^{(i)})^\top \phi^{(i)}$ where $\phi^{(i)} : \mathcal{S} \times \mathcal{A} \to [0, d_1]^{l^{(i)}}$ is an $l^{(i)}$-dimensional cost feature vector, $d_1$ is a bounded constant, and $\omega_E^{(i)} \in \mathbb{R}_+^{l^{(i)}}$ is the weight. The cost function of all the experts is $c_E \triangleq \sum_{i=1}^{N_E} c_E^{(i)}$. Expert $i$'s policy $\pi_E^{(i)}(a^{(i)}|s)$ represents the probability of expert $i$ taking action $a^{(i)}$ at state $s$ and the joint policy of all the experts is $\pi_E(a|s) \triangleq \prod_{i=1}^{N_E} \pi_E^{(i)}(a^{(i)}|s)$. We define $J_{r_E}(\pi) \triangleq E_{S,A}^\pi[\sum_{t=0}^T \gamma^t r_E(S_t, A_t)]$ as the expected cumulative reward under policy $\pi$ where the initial state is drawn from $P_0$ and $J_{c_E}(\pi) \triangleq E_{S,A}^\pi[\sum_{t=0}^T \gamma^t c_E(S_t, A_t)]$ as the expected cumulative cost. The experts' policy $\pi_E$ wants to maximize $J_{r_E}(\pi)$ subject to $J_{c_E}(\pi) \leq b$ where $b$ is the budget. Following [9, 18, 19], we study hard constraints (i.e., $b = 0$). These experts use $\pi_E$ to demonstrate $N_L$ trajectories $\{\zeta^{[v]}(n)\}_{v=1}^{N_L}$ at each online iteration $n$ where each trajectory $\zeta^{[v]}(n) \triangleq s_0^{[v]}(n), a_0^{[v]}(n), \cdots, s_T^{[v]}(n), a_T^{[v]}(n)$ is a state-action sequence of all the experts. This distributed online data style can also be found in [32].

**Learners**. There are $N_L$ learners where each learner $v$ knows $(\gamma, T, \{\phi^{(i)}\}_{i=1}^{N_E}, \zeta^{[v]}(n))$ and $\zeta^{[v]}(n)$ is a demonstration observed by learner $v$ at online iteration $n$. Each learner wants to use communications to learn the cost functions by estimating $\omega_E \triangleq [(\omega_E^{(1)})^\top, \cdots, (\omega_E^{(N_E)})^\top]^\top$ and reward functions using parameterized models $\{r_{\theta^{(i)}}^{(i)}\}_{i=1}^{N_E}$ where $\theta^{(i)} \in \mathbb{R}^{d^{(i)}}$ is $d^{(i)}$-dimensional. Here we relax the linear reward assumption in [9] but keep its linear cost assumption because non-linear cost functions can make the problem ill-defined (explained in Appendix).

**Assumption 1.** *The reward function $r_\theta$ satisfies the following: $|r_\theta(s, a)| \leq C$, $||\nabla_\theta r_\theta(s, a)|| \leq \bar{C}$, and $||\nabla_{\theta\theta}^2 r_\theta(s, a)|| \leq \tilde{C}$ for any $(s, a) \in \mathcal{S} \times \mathcal{A}$ and any $\theta$ where $C$, $\bar{C}$, and $\tilde{C}$ are positive constants.*

Notice that Assumption 1 is standard in RL [33, 34, 35].

The communication network is modeled as a time-varying directed graph $\mathcal{G}(n) \triangleq (\mathcal{V}, \mathcal{E}(n))$ where $\mathcal{V} \triangleq \{1, \cdots, N_L\}$ is the node (learner) set and $\mathcal{E}(n) \subseteq \mathcal{V} \times \mathcal{V}$ is the set of directed edges (communication links) at time $n$. The edge $(v, v') \in \mathcal{E}(n)$ means that learner $v$ receives information from learner $v'$ at time $n$ and $(v, v) \in \mathcal{E}(n)$ for all $n \geq 0$. The adjacency matrix of the graph at time $n$ is $W(n) \triangleq [W^{[vv']}(n)]_{v,v' \in \mathcal{V}} \in \mathbb{R}^{N_L \times N_L}$ where $W^{[vv']}(n) = 0$ if and only if $(v, v') \notin \mathcal{E}(n)$. The set of neighbors of learner $v$ at time $n$ is $\mathcal{N}^{[v]}(n) \triangleq \{v' \in \mathcal{V} | (v, v') \in \mathcal{E}(n)\}$.

**Assumption 2.** *There exists an integer $B \geq 1$ such that the graph $(\mathcal{V}, \mathcal{E}(n) \cup \cdots \cup \mathcal{E}(n + B - 1))$ is strongly connected for all $n \geq 0$.*

**Assumption 3.** *The adjacency matrix $W(n)$ has the following properties: (i) $\mathbf{1}^\top W(n) = \mathbf{1}^\top$ and $W(n)\mathbf{1} = \mathbf{1}$ where $\mathbf{1}$ is the column vector whose entries are all ones. (ii) There is an $\epsilon \in (0, 1)$ such that $W^{[vv]}(n) \geq \epsilon$ for all $v \in \mathcal{V}$ and $W^{[vv']}(n) \geq \epsilon$ if $(v, v') \in \mathcal{E}(n)$.*

Notice that these two assumptions are standard in distributed learning [3, 9, 36].

Figure 1 shows that the learners stand outside the MAS observing the sequential data revealed by the experts and collaboratively learn the reward functions and constraints. Each learner aims to learn the reward functions and constraints of all the experts where $\theta^{[v]}$ and $\omega^{[v]}$ are the reward and cost function parameter estimates of all the experts learned by learner $v$.

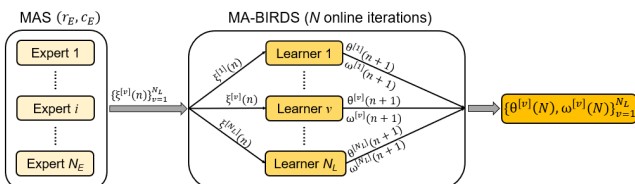

Figure 1: Relation between experts and learners

**Notions and Notations**. Define $\theta \triangleq [(\theta^{(1)})^\top, \cdots, (\theta^{(N_E)})^\top]^\top$ and $r_\theta \triangleq \sum_{i=1}^{N_E} r_{\theta^{(i)}}^{(i)}$. We use $\theta^{[v]}$ to represent learner $v$'s learned $\theta$. Therefore, the dimension of each $\theta^{[v]}$ is $\sum_{i=1}^{N_E} d^{(i)}$. Given a trajectory $\zeta = s_0, a_0, \cdots, s_T, a_T$, the empirical cumulative reward under $r_\theta$ is $\hat{J}_{r_\theta}(\zeta) \triangleq \sum_{t=0}^T \gamma^t r_\theta(s_t, a_t)$, the empirical cumulative cost feature is defined as $\hat{\mu}(\zeta) \triangleq \sum_{t=0}^T \gamma^t \phi(s_t, a_t)$ where $\phi \triangleq [(\phi^{(1)})^\top, \cdots, (\phi^{(N_E)})^\top]^\top$. The expectation of cumulative cost feature under a given policy $\pi$ is $\mu(\pi) \triangleq E_{S,A}^\pi[\sum_{t=0}^T \gamma^t \phi(S_t, A_t)]$ and the expectation of cumulative reward $r_\theta$ under a

given policy $\pi$ is $J_{r_\theta}(\pi) \triangleq E_{S,A}^\pi[\sum_{t=0}^T \gamma^t r_\theta(S_t, A_t)]$. The set of all stochastic policies is $\Pi$ where every $\pi \in \Pi$ satisfies $\pi(a|s) \geq 0$ for any $(s,a) \in \mathcal{S} \times \mathcal{A}$ and $\int_{a \in \mathcal{A}} \pi(a|s)da = 1$ for any $s \in \mathcal{S}$.

## 3  Problem Formulation

In MA-BIRDS, the learners collaboratively learn the experts' policy $\pi_E$ by estimating the reward function $r_E$ and cost function $c_E$, and each learner $v$ does not share its private data such as local trajectory $\zeta^{[v]}$ and local estimates $\hat{J}_{r_\theta}(\zeta^{[v]})$ and $\hat{\mu}(\zeta^{[v]})$. While it seems that learning a well-structured reward function is enough to prevent "bad" movements by assigning negative reward, we include a section (in Appendix) to further discuss the benefits of learning both reward and cost functions.

Many IRL works [9, 11, 24, 37? ] have a bi-level learning structure where the outer level is to learn a reward function and the inner level is to learn a corresponding policy by solving an RL problem under the current learned reward function. Inspired by their bi-level structure, we formulate a bi-level optimization problem where the outer level is to learn a reward function $r_\theta$ and the inner level is to find the cost function and policy corresponding to $r_\theta$. In what follows, we first define the inner-level optimization problem and then introduce the bi-level optimization problem.

**The inner-level optimization**. Given a learned reward function $r_\theta$, the corresponding policy $\pi_{r_\theta}$ is the optimal solution of the following constrained RL problem where the reward function is $r_\theta$:

$$\pi_{r_\theta} = \arg\max_{\pi \in \Pi} \left\{ H(\pi) + J_{r_\theta}(\pi), \quad \text{s.t. } \mu(\pi) = \frac{1}{N_L} \sum_{v=1}^{N_L} \hat{\mu}(\zeta^{[v]}(n)) \right\}, \tag{1}$$

where $H(\pi) \triangleq \sum_{t=0}^T E_{S,A}[-\gamma^t \ln \pi(A_t|S_t)]$ is causal entropy [38]. The constraint in problem (1) is cost feature expectation matching similar to the spirit of "feature expectation matching" in [11, 38].

However, the policy $\pi_{r_\theta}$ is hard to get because problem (1) is non-convex. A standard way to tackle this difficulty is dual methods [39], therefore, we introduce the dual function of problem (1): $G(\omega; \theta, n) \triangleq \max_{\pi \in \Pi} H(\pi) + J_{r_\theta}(\pi) + \omega^\top(\mu(\pi) - \frac{1}{N_L}\sum_{v=1}^{N_L}\hat{\mu}(\zeta^{[v]}(n)))$ where the dual variable $\omega$ is used to estimate $\omega_E$. Notice that the dual function $G(\omega; \theta, n)$ is convex in $\omega$ [39].

**Lemma 1.** *The optimal solution of problem* (1) *is the constrained soft Bellman policy $\pi_{\omega^*(\theta,n);\theta}$ and its parameter $\omega^*(\theta, n)$ is the optimal solution of the **dual problem** $\min_\omega G(\omega; \theta, n)$.*

The expression of constrained soft Bellman policy is in Appendix. Lemma 1 shows that $\pi_{\omega^*(\theta,n);\theta} = \pi_{r_\theta}$ and $\omega^*(\theta, n) = \arg\min_\omega G(\omega; \theta, n)$. Therefore, we can solve problem (1) by solving its dual problem. We use the dual problem to be the inner-level problem where $\omega^*(\theta, n)$ is the learned cost function and $\pi_{\omega^*(\theta,n);\theta}$ is the learned policy corresponding to the current learned reward function $r_\theta$.

**The bi-level optimization**. Given a reward function $r_\theta$, the inner-level problem $\arg\min_\omega G(\omega; \theta, n)$ can find the corresponding cost function and policy. The outer level aims to learn $r_\theta$ via minimizing the following loss function over $\theta$:

$$L(\theta, \omega^*(\theta, n), n), \quad \text{s.t. } \omega^*(\theta, n) = \arg\min_\omega G(\omega; \theta, n), \tag{2}$$

where $L(\theta, \omega^*(\theta, n), n) \triangleq -\sum_{v=1}^{N_L}\sum_{t=0}^T \gamma^t \ln \pi_{\omega^*(\theta,n);\theta}(a_t^{[v]}(n)|s_t^{[v]}(n))$ is the negative log likelihood of the trajectories $\{\zeta^{[v]}(n)\}_{v=1}^{N_L}$ received at time $n$ under the policy $\pi_{\omega^*(\theta,n);\theta}$ [9, 40] and $\zeta^{[v]}(n) = \{(s_t^{[v]}(n), a_t^{[v]}(n))\}_{0 \leq t \leq T}$. The likelihood function is widely used in IRL [9? , 40] to learn the reward function.

Notice that (2) requires all the demonstrations $\{\zeta^{[v]}(n)\}_{v=1}^{N_L}$ at time $n$. However, each learner $v$ can only observe $\zeta^{[v]}(n)$ and formulate its local negative log likelihood function $L^{[v]}(\theta, \omega^*(\theta, n), n) \triangleq -\sum_{t=0}^T \gamma^t \ln \pi_{\omega^*(\theta,n);\theta}(a_t^{[v]}(n)|s_t^{[v]}(n))$ and local dual function $G^{[v]}(\omega; \theta, n) \triangleq \max_{\pi \in \Pi} H(\pi) + J_{r_\theta}(\pi) + \omega^\top(\mu(\pi) - \hat{\mu}(\zeta^{[v]}(n)))$. Notice that $L = \sum_{v=1}^{N_L} L^{[v]}$ and $G = \frac{1}{N_L}\sum_{v=1}^{N_L} G^{[v]}$.

As the demonstrations are streaming, we have a sequence of loss functions $\{L(\cdot, \omega^*(\cdot, n), n)\}_{n \geq 1}$. We use this sequence of loss functions to formulate an online learning problem and a common problem for online learning is to minimize the regret: $\sum_{n=1}^N L(\theta(n), \omega^*(\theta(n), n), n) - \sum_{n=1}^N L(\theta^*, \omega^*(\theta^*, n), n)$

which quantifies the difference of the accumulative losses between the learned parameter $\theta$ and the best parameter $\theta^*$ in hindsight. However, it is too challenging to minimize the regret in our case because $L$ is non-convex [22]. Therefore, we use local regret [22, 23] which is widely used in online non-convex optimization. It quantifies the general stationarity of a sequence of loss functions. In specific, given a sequence of loss functions $\{f(\cdot, n)\}_{n \geq 1}$, the local regret [22, 23] at online iteration $n$ is defined as $||\frac{1}{l} \sum_{i=0}^{l-1} \nabla f(x(n), n-i)||^2$ which quantifies the gradient norms of the average of $l$ previously received loss functions under the current learned parameter $x(n)$. The total local regret is defined as the sum of the local regret at every online iteration $n$, i.e., $\sum_{n=1}^{N} ||\frac{1}{l} \sum_{i=0}^{l-1} \nabla f(x(n), n-i)||^2$. In our case, we replace $f$ with our loss function (2) and thereby formulate the local regret (3)-(4) which has a distributed bi-level formulation. We want to minimize the local regret (3)-(4).

$$\sum_{n=1}^{N} ||\frac{1}{l} \sum_{i=0}^{l-1} \sum_{v=1}^{N_L} \nabla L^{[v]}(\theta(n), \omega^*(\theta(n), n), n-i)||^2, \tag{3}$$

$$\text{s.t. } \omega^*(\theta(n), n) = \arg\min_{\omega} \sum_{v=1}^{N_L} G^{[v]}(\omega; \theta(n), n). \tag{4}$$

The time window length is $1 \leq l \leq N$ and $L^{[v]}(\theta, \omega, i) = 0$ if $i \leq 0$ [22]. The outer-level problem (3) is to learn the reward parameter $\theta$ and the inner-level problem (4) is to learn the cost parameter $\omega$ given $\theta$. The learned policy is the constrained soft Bellman policy $\pi_{\omega;\theta}$ with parameters $(\theta, \omega)$.

## 4 Algorithm and Performance Guarantee

This section consists of two subsections where the first one introduces an approximation method to solve the bi-level optimization problem (3)-(4) in a single loop and the second one introduces a consensus-based method for the multiple learners to solve the problem in a distributed way.

### 4.1 Approximation-based single-loop method

In this part, we develop an approximation-based single-loop method which (i) does not use the exact gradient of the outer-level problem (3) but an approximation of the gradient; (ii) solves the outer-level and inner-level problems in a single loop. In the following analysis, for simple notations, we omit the time index $n$ and imply that the analysis holds for all $n$.

**Lemma 2.** *The problem $\min_{\omega} G(\omega; \theta)$ has a unique optimal solution $\omega^*(\theta)$ for any $\theta$.*

Since the inner-level problem (4) is unconstrained and $\omega^*(\theta)$ is its optimal solution, then we have $\nabla_{\omega} G(\omega^*(\theta); \theta) = 0$. Taking derivative with respect to $\theta$ on both sides renders:

$$\nabla_{\omega\theta}^2 G(\omega^*(\theta); \theta) + \nabla_{\omega\omega}^2 G(\omega^*(\theta); \theta) \nabla \omega^*(\theta) = 0 \Rightarrow \nabla \omega^*(\theta) = -M(\theta, \omega^*(\theta))^\top,$$

where $M(\theta, \omega) \triangleq \nabla_{\theta\omega}^2 G(\omega; \theta)[\nabla_{\omega\omega}^2 G(\omega; \theta)]^{-1}$. Then, using the chain rule, we have:

$$\nabla L(\theta, \omega^*(\theta)) = \nabla_{\theta} L(\theta, \omega^*(\theta)) - M(\theta, \omega^*(\theta)) \nabla_{\omega} L(\theta, \omega^*(\theta)). \tag{5}$$

At online iteration $n$, an intuitive way to solve the bi-level optimization problem (3)-(4) is to solve it in a double-loop way [9, 26, 28]. In specific, the double-loop method first solves the inner-level problem (4) to find a close approximation of $\omega^*(\theta)$ and then uses the obtained result to get the gradient (5), thus solving the outer-level problem (3). However, it requires multiple gradient descent for the inner-level problem (4) to get $\omega^*(\theta)$ before updating the decision variable of the outer-level problem where each gradient descent of the inner-level problem needs to solve a constrained RL problem in our case. Therefore, the double-loop method is not suitable to be an online algorithm when the data is revealed in a fast speed. To design an algorithm suitable for the online fashion, we adopt the spirit of [29, 30] and solve the outer-level and inner-level problems in a single loop. However, our method is not a simple extension of [29, 30] to online settings because they are centralized and require the inner objective function to be strongly convex but our problem does not have these properties.

In specific, at each online iteration, the decision variables of both the outer-level and inner-level problems update only once where the gradient of the inner objective function is given in Lemma 3. For the outer-level problem, as $\omega^*(\theta)$ is inaccessible, we cannot get the exact gradient defined in

(5). Therefore, we propose the following gradient approximation whose derivation can be found in Appendix: $\bar{\nabla}L(\theta, \omega) = N_L E_{S,A}^{\pi_{\omega;\theta}}[\sum_{t=0}^{T} \gamma^t \nabla_\theta r_\theta(S_t, A_t)] - \sum_{v=1}^{N_L} \nabla_\theta \hat{J}_{r_\theta}(\zeta^{[v]})$, where $\pi_{\omega;\theta}$ is the constrained soft Bellman policy with parameters $(\omega, \theta)$. Compared to (5), this gradient approximation does not include the second-order term $M(\theta, \omega)$ and thus is much more computationally efficient. It is shown in Appendix that the approximation error $||\nabla L(\theta, \omega^*(\theta)) - \bar{\nabla}L(\theta, \omega)|| \leq C_\theta ||\omega^*(\theta) - \omega||$ where $C_\theta$ is a positive constant whose expression is in Appendix. Intuitively, as the inner decision variable $\omega$ approaches $\omega^*(\theta)$ in the learning process, this gradient approximation error can be sufficiently small.

**Lemma 3.** *The gradient of $G^{[v]}(\omega; \theta)$ is $\mu(\pi_{\omega;\theta}) - \hat{\mu}(\zeta^{[v]})$ where $\pi_{\omega;\theta}$ is the constrained soft Bellman policy with parameters $(\theta, \omega)$.*

**Lemma 4.** *The global likelihood function $L(\theta, \omega)$ is $C_L$-lipschitz continuous and $\bar{C}_L$-smooth in $(\theta, \omega)$, where $C_L$ and $\bar{C}_L$ are positive constants.*

### 4.2 Consensus-based distributed learning

In our distributed learning setting, each learner $v$ only knows its local information (e.g., $L^{[v]}$ and $G^{[v]}$), so that it cannot directly solve the problem (3)-(4). Therefore, each learner $v$ updates the decision variables using the gradients of its local outer-level and inner-level objective functions and uses communications to collaboratively solve the problem (3)-(4). Similar to the global gradient approximation $\bar{\nabla}L(\theta, \omega)$, learner $v$ has its local gradient approximation: $\bar{\nabla}L^{[v]}(\theta, \omega) = E_{S,A}^{\pi_{\omega;\theta}}[\sum_{t=0}^{T} \gamma^t \nabla_\theta r_\theta(S_t, A_t)] - \nabla_\theta \hat{J}_{r_\theta}(\zeta^{[v]})$. Learner $v$ uses samples to estimate $\bar{\nabla}L^{[v]}(\theta, \omega)$ via $\hat{\nabla}L^{[v]}(\theta, \omega) \triangleq \frac{1}{m^{[v]}} \sum_{j=1}^{m^{[v]}} \nabla_\theta \hat{J}_{r_\theta}(\zeta^j) - \nabla_\theta \hat{J}_{r_\theta}(\zeta^{[v]})$ where $\zeta^j$ is generated by rolling out $\pi_{\omega;\theta}$ and $m^{[v]}$ is the number of the samples. Moreover, learner $v$ also uses the same samples to estimate $\nabla G^{[v]}(\omega; \theta)$ in Lemma 3 via $\hat{\nabla}G^{[v]}(\omega; \theta) \triangleq \frac{1}{m^{[v]}} \sum_{j=1}^{m^{[v]}} \hat{\mu}(\zeta^j) - \hat{\mu}(\zeta^{[v]})$.

---

**Algorithm 1** Multi-agent behavior inference from distributed and streaming demonstrations

---

**Input**: $\{\theta^{[v]}(1)\}_{v=1}^{N_L}, \{\omega^{[v]}(1)\}_{v=1}^{N_L}, W(n)$
**Output**: $\theta^{[v]}(N), \omega^{[v]}(N), \pi_{\omega^{[v]}(N);\theta^{[v]}(N)}, \quad \forall v \in \mathcal{V}$
1: **for** $n = 1, \cdots, N$ **do**
2:     **for** $v \in \mathcal{V}$ **do**
3:         Finds policy $\pi_{\omega^{[v]}(n);\theta^{[v]}(n)}$ using soft Q-learning [41] or soft actor-critic [42].
4:         Simulates samples $\{\zeta^j\}_{j=1}^{m^{[v]}}$ using $\pi_{\omega^{[v]}(n);\theta^{[v]}(n)}$.
5:         Receives $\theta^{[v']}(n), \omega^{[v']}(n)$ from the neighbors $v' \in \mathcal{N}^{[v]}(n)$ and observes $\zeta^{[v]}(n)$.
6:         $\omega^{[v]}(n+1) = \sum_{v'=1}^{N_L} W^{[vv']}(n)\omega^{[v']}(n) - \frac{\alpha(n)}{l}\sum_{i=0}^{l-1} \hat{\nabla}G^{[v]}(\omega^{[v]}(n); \theta^{[v]}(n), n-i)$
7:         $\theta^{[v]}(n+1) = \sum_{v'=1}^{N_L} W^{[vv']}(n)\theta^{[v']}(n) - \frac{\beta(n)}{l}\sum_{i=0}^{l-1} \hat{\nabla}L^{[v]}(\theta^{[v]}(n), \omega^{[v]}(n), n-i)$
8:     **end for**
9: **end for**

---

In Algorithm 1, at online iteration $n$, each learner $v$ sequentially executes the following two steps: (i) uses current reward and cost function parameters to get the corresponding policy and generate samples (Lines 3-4); (ii) communicates with neighbors and updates its parameters for both outer-level and inner-level problems in a single loop using the samples generated in last step (Lines 5-7). In the update process (Lines 6-7), the first term (convex combination) encourages consensus among different learners and the second term (gradient) drives to the set of stationary points. Notice that even though soft Q-learning and soft actor-critic are designed for unconstrained RL, as shown in Appendix, we can revise them to approximate the constrained soft Bellman policy.

**Theorem 1.** *Suppose Assumptions 1, 2, 3 hold. Let the step sizes $\alpha(n) = \frac{\bar{\alpha}}{n^{\eta_1}}$ and $\beta(n) = \frac{\bar{\beta}}{n^{\eta_2}}$ where $\eta_1, \eta_2 \in (\frac{1}{2}, 1)$ and $\bar{\alpha}, \bar{\beta} \in (0, \frac{1}{N_L(2\bar{C}_L+1)})$, then for any $v, v' \in \mathcal{V}$ in Algorithm 1:*
*(consensus): $||\theta^{[v]}(N) - \theta^{[v']}(N)|| \leq O(\frac{1}{N^{\eta_1}} + \bar{\epsilon}^N)$, $||\omega^{[v]}(N) - \omega^{[v']}(N)|| \leq O(\frac{1}{N^{\eta_2}} + \bar{\epsilon}^N)$,*
*and $\sup_{(s,a)\in\mathcal{S}\times\mathcal{A}}\{|\pi_{\omega^{[v]}(N);\theta^{[v]}(N)}(a|s) - \pi_{\omega^{[v']}(N);\theta^{[v']}(N)}(a|s)|\} \leq O(\frac{1}{N^{\eta_1}} + \frac{1}{N^{\eta_2}} + \bar{\epsilon}^N))$, where $\sup_{(s,a)\in\mathcal{S}\times\mathcal{A}}\{\cdot\}$ outputs the supremum value over $\mathcal{S}\times\mathcal{A}$ and the expression of $\bar{\epsilon} \in (0, 1)$ can be*

*found in Appendix.*

*(decreasing average local regret):* $\frac{1}{N}\sum_{n=1}^{N} E\big[\frac{1}{l}\|\sum_{i=0}^{l-1}\nabla L(\theta^{[v]}(n),\omega^{[v]}(n),n-i)\|^2\big] \leq O(\frac{1}{N^{1-\eta_1}}+\frac{1}{N^{1-\eta_2}}+\frac{1}{N})+\frac{2(C_L)^2}{l}.$

*(sub-linear cumulative constraint violation):* $\sum_{n=1}^{N} E\big[J_{c_E}^2(\pi_{\omega^{[v]}(n);\theta^{[v]}(n)})\big] \leq O(N^{\eta_2}+1)$, *where the expectation is taken over randomly demonstrated trajectories at each online iteration.*

Theorem 1 shows that the learners will achieve consensus on the reward and cost function parameters, and policy. The average local regret decreases to an upper bound $2(C_L)^2/l$. Notice that this upper bound has similar property with that in [22, 23]: (i) they are both less that the uniform upper bound $C_L^2$ of the global loss function $L$ when $l > 2$; (ii) they are both diminishing if $l$ is dependent on $n$ and $l(n) = \omega(1)$ [23]. The number of total outer gradient steps taken by each learner in Algorithm 1 is $O(N)$ which is smaller than $O(2Nl + l^2)$ in [22] and $O(Nl^2)$ in [23]. The reason of fewer gradient steps in Algorithm 1 is that it is an online gradient descent algorithm, which does not require multiple gradient calls at each online iteration as follow-the-leader-based methods [22, 23] do.

Moreover, if the reward functions are linear as in [20, 21], we have the following stronger result:

**Corollary 1.** *If the reward functions are linear, the average difference of cumulative reward between the learned policy and expert policy diminishes:* $\frac{1}{N}\sum_{n=1}^{N} E\big[(J_{r_E}(\pi_{\omega^{[v]}(n);\theta^{[v]}(n)})-J_{r_E}(\pi_E))^2\big] \leq O(\frac{1}{N^{1-\eta_1}}+\frac{1}{N})$ *for any* $v \in \mathcal{V}$.

## 5   Simulations

This section shows that Algorithm 1 is effective to both discrete and continuous environments. We use four centralized baselines for comparisons: (i) **Behavior inference from centralized and streaming demonstrations (BICS)**: This is the centralized counterpart of MA-BIRDS where a central learner obtains all the demonstrations at each online iteration. (ii) **Follow the leader (FTL)**: This method [22] uses a follow-the-leader-based method to solve online non-convex optimization problems where a (near)-stationary solution of the sum of all the previous loss functions is found at each online iteration. As this method only solves single-level optimization problems, we revise it to a single-loop method to solve bi-level optimization problems. (iii) **Double-loop method (DLM)**: This method extends D-ICRL [9] to online centralized settings where the inner-level problem is first solved and then the result is used to solve the outer-level problem. At each online iteration, DLM only updates the decision variable of the outer-level problem once. (iv) **ME-greedy**: This method is an online extension of [18] which assumes the access to the ground truth reward and uses a greedy method to estimate the constraints based on maximum entropy (ME) IRL [11].

### 5.1   Evasion from patrolled area

We consider the evader-patroller setting (Figure 2a) introduced in [21]. The experts (E1 and E2) are the patrollers patrolling around the area and they aim to switch their positions. They have four actions (i.e., moving up, down, left, and right) and need to avoid collisions with each other and the obstacles (i.e., constraints). The experts are programmed to follow the optimal policy. The learners (L1 and L2) are the evaders who want to learn the behavior model of the experts in order to reach the goal G without being caught. The red crosses in Figure 2 represent the constraints and each state is colored according to the scaled visitation frequency.

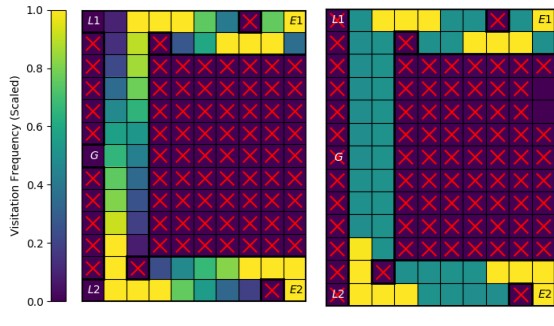

(a) Ground truth environment   (b) Learned environment

Figure 2: Evader-patroller environment.

Figure 2b shows the results learned by Algorithm 1 (each learner recovers the same constraints). Two constraints are not recovered as the experts' policy will not change even these constraints are absent. Three constraints are falsely learned because the experts (optimal policy) do not visit those states but the learned policy (constraint soft Bellman policy) have a high chance of visiting the states if the

states are not prohibited as the constrained soft Bellman policy has non-zero probability of choosing any action and this probability can be large if the corresponding action is not heavily penalized.

To reason about the performance of our algorithm, we use four metrics: false positive rate (FPR), false negative rate (FNR), constraint violation rate (CVR), and success rate (SR). The FPR [18], is the proportion of learned constraints that are not the ground truth constraints, FNR is the proportion of the ground truth constraints that are not learned, CVR introduced in [19] is the percentage of the learned policy violating any constraint, and SR is the percentage of the learned policy reaching the destinations and avoiding collisions.

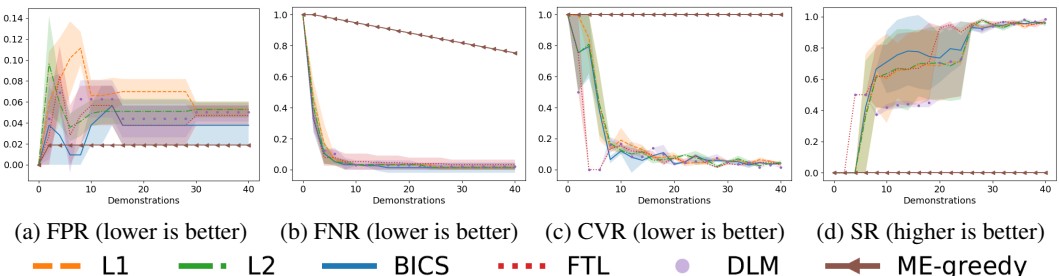

(a) FPR (lower is better)  (b) FNR (lower is better)  (c) CVR (lower is better)  (d) SR (higher is better)

— — L1   — · L2   —— BICS   ···· FTL   • DLM   ◄—— ME-greedy

Figure 3: Algorithm performance. The L1 and L2 are the distributed learners in MA-BIRDS.

Figure 3 shows that Algorithm 1 is on par with and even outperforms the baselines even if it is distributed and does not have the access to the ground truth reward. For the baselines, the centralized learners receive two demonstrations at each online iteration. We can see that the gradient-based methods (i.e., MA-BIRDS, BICS, FTL, and DLM) have much better performance than the greedy-based method (i.e., ME-greedy). The reason is that the greedy-based method can only learn one constraint at each online iteration while the gradient-based methods can learn multiple constraints at each online iteration because the gradient-based methods update $\omega$ which works on all the possible constraints indicated by the cost feature vector $\phi$.

Table 1: Performance comparisons. Here, D means distributed and NATR means no access to the (ground truth) rewards.

|  |  | D | NATR | FPR | FNR | CVR | SR |
|---|---|---|---|---|---|---|---|
| MA- | L1 | ✓ | ✓ | $0.053 \pm 0.008$ | $0.013 \pm 0.013$ | $0.040 \pm 0.010$ | $0.960 \pm 0.010$ |
| BIRDS | L2 | ✓ | ✓ | $0.053 \pm 0.008$ | $0.013 \pm 0.013$ | $0.045 \pm 0.015$ | $0.955 \pm 0.015$ |
| BICS |  | × | ✓ | $0.038 \pm 0.019$ | $0.013 \pm 0.013$ | $0.040 \pm 0.005$ | $0.960 \pm 0.005$ |
| FTL |  | × | ✓ | $0.047 \pm 0.009$ | $0.032 \pm 0.032$ | $0.030 \pm 0.010$ | $0.970 \pm 0.010$ |
| DLM |  | × | ✓ | $0.050 \pm 0.009$ | $0.022 \pm 0.022$ | $0.015 \pm 0.005$ | $0.985 \pm 0.005$ |
| ME-greedy |  | × | × | $0.019 \pm 0.000$ | $0.753 \pm 0.000$ | $1.000 \pm 0.000$ | $0.000 \pm 0.000$ |

Table 1 shows the final results after the 40 demonstrations are revealed. Notice that even if the gradient-based methods have different results in FPR and FNR, they achieve similar performance in CVR and SR. The reason is that the different constraints they learn will not affect the learned policy as those constraints are either blocked by other constraints or occupy the states that the experts will barely visit (as shown in Figure 2b).

Moreover, to show that MA-BIRDS is suitable for online learning when the streaming data arrives at a fast speed, we propose the following three metrics: scaled time per online iteration (STPOI), 50% SR scaled time (50% ST), and 90% SR scaled time (90% ST). The STPOI is the scaled time that an algorithm needs to finish the computation of an online iteration, 50% ST is the scaled time that an algorithm needs to reach 50% SR, and 90% ST is the scaled time to reach 90% SR. We use scaled time instead of actual time because actual time varies a lot on different

Table 2: Computation time comparisons.

|  | STPOI | 50% ST | 90% ST |
|---|---|---|---|
| L1 | 1.000 | 1.000 | 1.000 |
| L2 | 1.000 | 1.000 | 1.000 |
| BICS | 1.157 | 1.155 | 1.158 |
| FTL | 7.500 | 3.753 | 5.769 |
| DLM | 7.482 | 3.740 | 7.481 |
| ME-greedy | 63.615 | > 318.075 | > 97.869 |

hardwares and different problems. The time is scaled in the way that the fastest algorithm has scaled time 1.000.

Table 2 shows that MA-BIRDS and BICS are much faster than the baselines in each online iteration, and thus are more suitable for the online setting with fast streaming data. The MA-BIRDS is slightly faster than BICS in each online iteration because the centralized learner needs to process two new demonstrations at each online iteration while each distributed learner only needs to process one. Moreover, we can see that MA-BIRDS achieves 50% and 90% SR with the shortest time.

## 5.2 Drone motion planning with obstacles

In this example, we analyze MA-BIRDS on a real-world problem with continuous state and action spaces introduced in [9]. We build a simulator in Gazebo (Figure 4a) where we (humans) control the two drones to their diagonal doors while avoiding collisions. We reveal each of the four learners one demonstration at each online iteration. In this experiment, the learners are the computers recording the demonstrations of the experts (humans) controlling the drones. As the state and action spaces are continuous, the potential constraints can also be continuous. Therefore, we cannot use the metrics FPR and FNR, and we cannot use the baseline ME-greedy. As ME-greedy is designed for discrete state-action space, we replace it with ME-gradient [19] which extends ME-greedy to

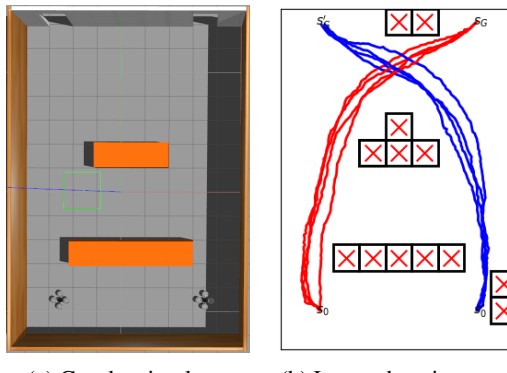

(a) Gazebo simulator     (b) Learned environment

Figure 4: Drone motion planning with obstacles.

continuous settings where a cost function is learned using a gradient-based method. Similar to ME-greedy, ME-gradient is centralized and assumes access to the ground truth reward. Figure 4b shows the learned constraints on which the learners reach consensus and the trajectories of the learned policy. Each of the four trajectories is generated by one of the four learners L1-L4. Each trajectory consists of the paths of the two drones (i.e., a red path and a blue path).

Table 3: Performance and computation time comparisons (drone motion planning).

| | D | NATR | CVR | SR | STPOI | 50% ST | 90% ST |
|---|---|---|---|---|---|---|---|
| L1 | ✓ | ✓ | $0.025 \pm 0.025$ | $0.974 \pm 0.025$ | 1.000 | 1.000 | 1.000 |
| L2 | ✓ | ✓ | $0.030 \pm 0.010$ | $0.970 \pm 0.010$ | 1.000 | 1.000 | 1.000 |
| L3 | ✓ | ✓ | $0.025 \pm 0.025$ | $0.973 \pm 0.025$ | 1.000 | 1.000 | 1.000 |
| L4 | ✓ | ✓ | $0.025 \pm 0.025$ | $0.972 \pm 0.015$ | 1.000 | 1.000 | 1.000 |
| BICS | ✗ | ✓ | $0.015 \pm 0.015$ | $0.979 \pm 0.020$ | 1.029 | 1.025 | 1.027 |
| FTL | ✗ | ✓ | $0.015 \pm 0.010$ | $0.981 \pm 0.018$ | 10.481 | 7.860 | 5.240 |
| DLM | ✗ | ✓ | $0.020 \pm 0.015$ | $0.975 \pm 0.015$ | 9.658 | 6.767 | 6.837 |
| ME-gradient | ✗ | ✗ | $0.013 \pm 0.013$ | $0.980 \pm 0.018$ | 1.013 | 1.010 | 1.012 |

Table 3, together with Table 2, shows that MA-BIRDS and BICS can reach the same good performance with the baselines and use much shorter time or have fewer requirements. In specific, compared to FTL [22], MA-BIRDS only requires about 20% of the time to reach the same performance and is more than six times faster in each iteration. Compared to DLM [9], MA-BIRDS only requires about 15% of the time to reach the same performance and is also more than six times faster in each iteration. Compared to ME-gradient, MA-BIRDS can reach the same performance within the same amount of time even if it is distributed and has no access to the ground truth reward.

## 6 Discussion and future work

We propose MA-BIRDS, the first IRL framework that is effective in learning multi-agent behaviors from distributed and streaming demonstrations under continuous and discrete environments. We formulate a distributed online bi-level optimization problem and propose a fast distributed online gradient descent single-loop algorithm with theoretical guarantees that is suitable to online settings.

Experimental results show that MA-BIRDS is effective in both continuous and discrete environments. Despite its benefits, one limitation is the assumption of linear cost functions. We will explore approaches to relax this assumption in the future.

**Potential negative social impact**. Since MA-BIRDS can infer the experts' behaviors, potential negative social impact may occur if the learners are malicious. Take the evader-patroller setting as an example; the patrollers could be safeguards or park rangers, and the evaders could be poachers. The malicious poachers may use MA-BIRDS to escape. To avoid this situation, the experts should take additional strategies such as regularly demonstrating misleading behaviors so that MA-BIRDS will learn a wrong behavior model of the experts.

# 7 Acknowledgements

This work is partially supported by the National Science Foundation through grants ECCS 1846706 and CNS 1830390. We would like to thank the reviewers for their insightful and constructive suggestions.

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
