# OpenReview forum: "Learning Multi-agent Behaviors from Distributed and Streaming Demonstrations"
_NeurIPS.cc/2023/Conference — NeurIPS 2023 poster_

### Official Review · Reviewer_9ZLE · 2023-06-30

**Soundness:** 3 good
**Presentation:** 3 good
**Contribution:** 3 good
**Rating:** 7
**Confidence:** 3

**Summary:**

The authors study the problem of uncovering the incentives and behaviors of a cooperative multiagent system (MAS) that is optimizing for an objective subject to constraints. The authors consider solving this as a distributed optimization problem where several learners observe separate trajectories of the MAS in parallel and infer the parameters of the MASs reward functions and policies. The learners periodically pool this information to arrive at a consensus. This problem is posed as a bi-level optimization problem where the inner problem of recovering a joint policy given a reward function is solved in the dual, and the outer problem is to learn the reward function via maximum likelihood estimation. The authors prove a consensus will be reached and give a convergence rate. They also prove the agreed upon policy enjoys sublinear regret as measured by constraint violation. The authors demonstrate their approach on two domains, a discrete evader-patroller domain and a continuous drone domain. MA-BIRDS is on par with prior centralized approaches (e.g, BICS) in terms of uncovering correct constraints and high performing policies but with slightly shorter runtimes.

**Strengths:**

Given the complexity of the setting, I thought the authors did an excellent job explaining their setup and solution clearly. They clearly acknowledge prior work and explain their contribution to inferring the behaviors of an MAS with a *distributed*, *online* approach. They support their algorithmic development with theoretical analysis and run experiments comparing against several strong baselines.

**Weaknesses:**

I feel the paper could better motivate the *distributed* problem. The experiments show centralized approaches are nearly as fast and with similar, if not better performance. Distributed approaches can often take advantage of parallelism to achieve faster convergence at scale, but I did not see that reflected in the theoretical rates, nor did I see experiments testing this.

**Questions:**

- Can you give intuition for why your theoretical convergence rates are independent of $N_L$?
- One of the benefits of distributed learning is the ability to increase parallel computation to achieve faster learning. Did you vary $N_L$ at all for either experiment and can you include those results if you have them?

**Limitations:**

The evader-patroller setting suggests the proposed method could be used in a way that has negative societal impact. Typically, we want the patrollers (e.g., park rangers) to capture the evaders (e.g., poachers). Can you discuss this briefly?

---

> ### Author Rebuttal · Authors · 2023-08-08
>
> Thanks for your constructive reviews. We believe that this discussion will help improve the paper. We address your comments below:
>
> **Weakness: I feel the paper could better motivate the distributed problem. The experiments show centralized approaches are nearly as fast and with similar, if not better performance. Distributed approaches can often take advantage of parallelism to achieve faster convergence at scale, but I did not see that reflected in the theoretical rates, nor did I see experiments testing this.**
>
> **Answer**: Thanks for mentioning the parallelism. The parallelism in our case may not significantly accelerate the learning. The reason is that the distributed learners only share the burden of data processing, which only consumes a small portion of the total computation time. Note that solving the RL problem (line 3 in Algorithm 1) consumes most of the computation time. Hence, distributed learning in our paper is not significantly faster than centralized learning because both learning methods need to solve an RL problem at each online iteration. This is consistent with the empirical results where the distributed learners are only slightly faster than the centralized learner at each online iteration.
>
> We agree that parallelism can accelerate learning in some cases. For example, deep learning over a very large set of images [D1] where the global dataset is distributed to many local datasets and each learner only processes local data, computes the gradient using local data, and updates parameters via gradient descent in parallel. This kind of parallelism can significantly accelerate the learning because the major burden, i.e., processing the data and computing gradients using the data, is shared by the distributed learners. However, in our case, the major burden is to solve the RL problem and both distributed learning and centralized learning suffer the issue. Therefore, the parallelism cannot significantly accelerate the learning in our case.
>
> Mentioned in lines 15-21, the data of multi-agent systems is usually distributed and thus centralized learning is infeasible. Therefore, the distributed learning in our case is not to accelerate the learning process but as a good substitute for the centralized learning since the distributed learning enables multiple learners to collaboratively solve the global learning problem over ad hoc networks without a centralized authority while each learner only knows a portion of training data.
>
> **Q1:  Can you give intuition for why your theoretical convergence rates are independent of $N_L$?**
>
> **Answer**: Thank you for pointing this out. Our convergence actually depends on $N_L$. We omit the dependence of $N_L$ in the theorem statement because we treat it as a constant with respect to the iteration number $N$. We will add this dependency for better understanding. The conclusion is that the convergence will be slower if $N_L$ becomes larger. The intuition behind this is that a larger number of learners will make it harder to achieve consensus, thus resulting in a slower convergence. Note that the distributed learning can be slightly faster at each online iteration, however, the convergence rate is to quantify the iteration complexity and a larger $N_L$ may result in more iterations to converge. The theoretical reason of slower convergence is in Appendix. In specific, from line 210 in Appendix, the convergence depends on the constants $C_1,C_2,C_3$ and the convergence will become slower for larger $C_1,C_2,C_3$. From lines 207-209 in Appendix, we see that $C_1,C_2,C_3$ will become larger if the constants $C\_{max},\tilde{C}\_{max}$ become larger. From Lemma 10 and line 179 in Appendix, we can see that $C\_{max}$ becomes larger if $B_0$ becomes larger. From lines 199-205, we can see that $\tilde{C}\_{max}$ becomes larger if $B_0$ becomes larger. From line 155 in Appendix, we know that $B_0=(N_L-1)B$ where $B$ is a positive constant defined in Assumption 2. Therefore, a larger $N_L$ will eventually lead to larger $C_1,C_2,C_3$, and thus slower convergence.
>
> **Q2: One of the benefits of distributed learning is the ability to increase parallel computation to achieve faster learning. Did you vary $N_L$ at all for either experiment and can you include those results if you have them?**
>
> **Answer**: The parallel computation among the distributed learners will not significantly accelerate the learning in our case. Please refer to the answer to the weakness for the reasons. We include a table (i.e., Table 1) in the newly uploaded PDF file to compare the scaled computation time for different numbers of learners. From the table, we can see that the computation time at each online iteration remains almost unchanged for different numbers of learners. However, the time for convergence, e.g., reaching certain performance, becomes longer when the number of learners increases. The reason is that it takes more iterations to reach consensus for a larger number of learners.
>
> **Limitation: The evader-patroller setting suggests the proposed method could be used in a way that has negative societal impact. Typically, we want the patrollers (e.g., park rangers) to capture the evaders (e.g., poachers). Can you discuss this briefly?**
>
> **Answer**: We appreciate that you mention the potential negative social impact. We would like to add the following discussion:
>
> Since MA-BIRDS can infer the experts' behaviors, potential negative social impact may occur if the learners are malicious. Take the evader-patroller setting as an example, the patrollers could be safeguards or park rangers and the evaders could be poachers. The malicious poachers may use MA-BIRDS to escape. To avoid this situation, the experts should take additional strategies such as regularly demonstrate misleading behaviors so that MA-BIRDS will learn a wrong behavior model of the experts.
>
> [D1] V. Hegde and S. Usmani, "Parallel and distributed deep learning," Stanford University, 2016.

---

> > ### Comment · Reviewer_9ZLE · 2023-08-11
> > **Acknowledgement of Rebuttal**
> >
> > Dear authors, thank you for your rebuttal. The example you gave regarding potential negative societal impact is perfect, thank you. And thank you for your discussion on parallelism and the motivation for distributed learning. Could you expand on this motivation further? Even if multiagent demonstrations are generated and naturally stored in a distributed way, what are the downsides to the naive approach of first centralizing the storage and then running the centralized solver? Is the data so large that it wouldn't fit in centralized storage? Would centralization be slow? Are there privacy concerns regarding the demonstrations? etc. Thank you.

---

> > > ### Author Response · Authors · 2023-08-12
> > > **Response to the Reviewer**
> > >
> > > Dear reviewer,
> > >
> > > Thanks for mentioning the method of first centralizing the data and then running centralized solver.
> > >
> > > However, this method has several potential issues. First, sharing distributed raw demonstration data could breach the privacy of data owners. For example, in a self-driving system [D2], each autonomous vehicle has its own demonstration data, and they are not willing to share some raw demonstration data, e.g., GPS location, because those data may contain the user's private information, e.g., user’s location and common destinations. This can potentially reveal the information of where a person lives, works, and shops, thus helping infer the information about income level and spending habits [D3].
> > >
> > > Second, the communication overhead could be high if all the distributed data is sent to a centralized solver, especially when the data is of high dimension. Consider that a machine learning model is trained over a massive number of videos where each distributed device (e.g., mobile phone or individual computer) contains a certain number of videos. For example, [D4] uses distributed learning to train over the video dataset Kinetics (438GB). Suppose the distributed learners need to communicate all these videos to the centralized solver, the communication burden is at least 438GB. In contrast, in the distributed learning, the distributed learners do not share the raw data (i.e., the videos) but only the parameters of the trained model (24.3MB in [D4]) which are of much lower dimension.
> > >
> > > Third, centralized learning is not robust to failures because the whole learning process will terminate if the centralized learner fails. In contrast, distributed learning can still operate in a degradation mode even if one or more individual learners fail [D5].
> > >
> > > Fourth, in some cases, the data could be too large and complicated to move and centralize such as the astronomical data [D6] where each local dataset could be terabytes in size and contain hundreds of millions of objects separated into millions of files. It is infeasible to centralize multiple such local datasets.
> > >
> > > [D2] D. Glancy, "Privacy in autonomous vehicles," Santa Clara Law Review, vol. 52, no. 4, p. 1171, 2012.
> > >
> > > [D3] P. Pype, G. Daalderop, E. Schulz-Kamm, E. Walters, and M. von Grafenstein, “Privacy and security in autonomous vehicles,” Automated Driving: Safer and More Efficient Future Driving, pp.17-27, 2017.
> > >
> > > [D4] J. Lin, C. Gan, S. Han, "Training Kinetics in 15 mins: Large-scale distributed training on videos," arXiv, preprint arXiv: 1910.00932, 2019.
> > >
> > > [D5] N. Vlassis, “A concise introduction to multi-agent systems and distributed artificial intelligence,” Synthesis Lectures on Artificial Intelligence and Machine Learning, vol. 1, no. 1, pp. 1-71, 2007.
> > >
> > > [D6] I. Raicu, I. Foster, A. Szalay, and G. Turcu, “Astroportal: A science gateway for large-scale astronomy data analysis,” Teragrid Conference, pp. 12-15, 2006.
> > >
> > > Best regards,
> > >
> > > Authors.

---

> > > > ### Comment · Reviewer_9ZLE · 2023-08-15
> > > >
> > > > Thank you for your explanation. I think adding this kind of discussion to the paper would help better motivate the distributed setting.
> > > >
> > > > If possible, it would also be helpful to consider modelling one of these four cases (privacy, limited comm bandwidth, node failures, memory storage limits) in one of your experiments to support your arguments above.

---

> > > > > ### Author Response · Authors · 2023-08-16
> > > > > **Response to the reviewer**
> > > > >
> > > > > Dear reviewer,
> > > > >
> > > > > Thanks for your constructive advice. We will add this discussion to the paper for a better motivation of the distributed learning. We will also present the communication burden comparison for the drone experiment to show that the distributed learning has much lower communication burden compared to the naive centralized method which gathers all the distributed demonstrations. In specific, let us first consider the communication burden at each online iteration. At each online iteration, four pairs of new trajectories are demonstrated to the four learners and each trajectory has a length of 500, i.e., 500 joint state-action pairs. Each joint state-action pair has the dimension of $8$ and thus each joint state-action pair is $8\times8=64$ bytes given that we use "double" as the datatype. The total size of one demonstrated trajectory is $500\times64=32,000$ bytes. If we gather all the four trajectories in a centralized entity, the total communication burden at each online iteration is $4\times32,000=128,000$ bytes. In distributed learning, each distributed learner only communicates its parameters once at each online iteration. The parameters have the dimension of $512$. Therefore, the communication burden of each learner is $512\times8=4096$ bytes and the total communication burden of all the learners is $4\times4096=16,384$ bytes at each online iteration which is only about one eighth of that of the naive centralized method. Note that the centralized method needs to gather four new pairs of trajectories at each online iteration. If we consider the communication burden over the whole learning process, the communication burden of the distributed learning is still one eighth of the centralized method because the total communication burden of both methods is the communication burden of each online iteration multiplying the number of online iterations. We will add such experiment results to better motivate the distributed learning.
> > > > >
> > > > >  Best regards,
> > > > >
> > > > > Authors.

---

> > > > > > ### Comment · Reviewer_9ZLE · 2023-08-21
> > > > > >
> > > > > > Thank you for your detailed example. I think this would be a valuable addition to the appendix.

---

### Official Review · Reviewer_QLML · 2023-07-08

**Soundness:** 3 good
**Presentation:** 3 good
**Contribution:** 2 fair
**Rating:** 5
**Confidence:** 4

**Summary:**

A new type of the MA-IRL problem is proposed and analysed in this paper, which features online learning, distributed demonstrations and constraints independent of rewards. To tackle challenges resulting from these features, the authors design a novel algorithm called MA-BIRDS that utilises a single-loop mechanism to learn the underlying reward functions, constraints and policies. Due to the decentralised nature, the local regret is used to approximate the actual gradient. Authors upper bound the local regret at an ideal case and provide a theoretical guarantee on the achievement of a consensus over the learned rewards, policy and costs among all agents. Fruitful experiments are executed that validate the effectiveness of the proposed algorithm.

**Strengths:**

The paper contributes a novel IRL algorithm that is adapted to streaming data and distributed multi-agent systems. The single-loop update of rewards and policies sets this work apart from the others in the literature in the sense that it can considerably alleviate the time pressure and is hence suitable for online scenarios. Theoretical results on convergence and consistency pin the proposed algorithm deeply down to reliability.

**Weaknesses:**

Although I agree that the single-loop update of rewards and policies is a significant contribution to the literature of IRL, I still doubt the online efficiency of the proposed algorithm because the soft-Q learning/soft actor-critic is involved in each round of updates. As far as I know, soft-Q learning or soft actor-critic is not a kind of fast-converging RL algorithm, or at least they cannot always guarantee fast convergence. This would potentially bring some negative effects on online efficiency when the problem scale is large or the online data comes with a complex structure. I would like to increase the score if the authors can make a discussion on this and/or provide some empirical evidence to clarify this.

**Questions:**

N/A

---

> ### Author Rebuttal · Authors · 2023-08-08
>
> Thanks for your constructive and insightful review. Discussing this point will help us improve and clarify our work. We address the comment below.
>
> **Weakness: Although I agree that the single-loop update of rewards and policies is a significant contribution to the literature of IRL, I still doubt the online efficiency of the proposed algorithm because the soft-Q learning/soft actor-critic is involved in each round of updates. As far as I know, soft-Q learning or soft actor-critic is not a kind of fast-converging RL algorithm, or at least they cannot always guarantee fast convergence. This would potentially bring some negative effects on online efficiency when the problem scale is large or the online data comes with a complex structure. I would like to increase the score if the authors can make a discussion on this and/or provide some empirical evidence to clarify this.**
>
> **Answer**: It is very insightful of you to point out that the reinforcement learning step (e.g., soft Q learning or soft actor-critic) can be slow, thus resulting in a slow reward and cost update at each online iteration. This is indeed a problem for many IRL [C1,C2,C3] and online IRL [C4,C5] works since it is typical to require to solve an RL problem at each iteration for reward update. However, this problem can be alleviated by two ways.
>
> The first one is warm start. In specific, at each online iteration, we need to compute the policy corresponding to the new reward and cost functions. Instead of computing the policy from random initialization, we use the results of the last online iteration as the initialization of this online iteration. Take soft actor-critic as an example, we can use the learned parameters of the policy net and value net at last online iteration as the initialization of the policy net and value net at current online iteration. Intuitively, if the reward and cost functions between two consecutive online iterations are close, it is expected that the corresponding policy and value nets should have similar parameter values. Therefore, the warm start can speed up the convergence of RL since it provides a good initialization. Moreover, The distance of reward parameters and the distance of cost parameters between two consecutive online iterations are indeed close in our situation. Since the step sizes $\alpha(n)$ and $\beta(n)$ are diminishing and the gradient of $L$ and $G$ are bounded (proved in Lemma 8 and lines 148-149 in Appendix), the distance of reward parameters and the distance of cost parameters between any consecutive online iterations will diminish. Therefore, the RL step will become faster and faster when the online iteration index $n$ increases. We include a figure (Figure 2) in the newly uploaded PDF file to show the scaled computation time of each online iteration for the evader-patroller experiment. From the figure, we can see that the warm start can significantly accelerate the RL process.
>
> The second one is to use only one-step policy update at each online iteration. This idea has been proved to be effective in the offline setting [C6]. In specific, given the new reward and cost function at each online iteration, we only perform one-step policy update from the policy in the last online iteration instead of computing the corresponding constrained soft Bellman policy. The one-step policy update could be the policy update in soft Q learning or actor update in soft actor-critic. Since we only update the policy by one step instead of fully solving an RL problem, it will be much faster. Theoretically, this idea can work because of two-timescale stochastic approximation where the policy updates in a faster timescale and the reward and cost update in a slower timescale. The policy update is faster because it converges linearly under fixed reward and cost functions [C7] while the reward and cost updates are slower given that $\alpha(n)\propto n^{-\eta_1}$ and $\beta(n)\propto n^{-\eta_2}$ where $\eta_1,\eta_2\in(\frac{1}{2},1)$. Intuitively, since the policy update is faster than the reward and cost updates, the reward and cost parameters are "relatively fixed" compared to the policy. It is expected that the policy shall stay close to the corresponding constrained soft Bellman policy and at last converges to the corresponding constrained soft Bellman policy when $n$ increases.
>
> [C1] P. Abbeel and A. Y. Ng, “Apprenticeship learning via inverse reinforcement learning,” International Conference on Machine Learning, pp. 1–8, 2004.
>
> [C2] B. D. Ziebart, A. L. Maas, J. A. Bagnell, and A. K. Dey, "Maximum entropy inverse reinforcement learning,” National Conference on Artificial Intelligence, pp. 1433–1438, 2008.
>
> [C3] S. Arora and P. Doshi, “A survey of inverse reinforcement learning: Challenges, methods and progress,” Artificial Intelligence, vol. 297, p. 103500, 2021.
>
> [C4] N. Rhinehart and K. M. Kitani, “First-person activity forecasting with online inverse reinforcement learning,” IEEE International Conference on Computer Vision, pp. 3696–3705, 2017.
>
> [C5] S. Arora, P. Doshi, and B. Banerjee, “Online inverse reinforcement learning under occlusion,”
> International Conference on Autonomous Agents and Multiagent Systems, pp. 1170–1178, 2019.
>
> [C6] S. Zeng, C. Li, A. Garcia, and M. Hong, “Maximum-likelihood inverse reinforcement learning with finite-time guarantees,” Advances in Neural Information Processing Systems, pp. 10122-10135, 2022.
>
> [C7] S. Cen, C. Cheng, Y. Chen, Y. Wei, and Y. Chi, "Fast global convergence of natural policy gradient methods with entropy regularization," Operations Research, vol. 70, no. 4, pp. 2563–2578, 2022

---

> > ### Comment · Reviewer_QLML · 2023-08-19
> >
> > Thank you for the response. I am more inclined to maintain the score.

---

### Official Review · Reviewer_grYB · 2023-07-09

**Soundness:** 3 good
**Presentation:** 2 fair
**Contribution:** 3 good
**Rating:** 5
**Confidence:** 3

**Summary:**

The paper introduces MA-BIRDS, an Inverse Reinforcement Learning (IRL) framework designed to effectively learn multi-agent behaviors from distributed and streaming demonstrations in continuous and discrete environments. The authors initially frame the problem as a distributed online bi-level optimization problem. They subsequently present a fast distributed online gradient descent single-loop algorithm, accompanied by theoretical guarantees. To showcase the efficacy of MA-BIRDS, the paper includes a set of experiments conducted in both continuous and discrete environments.

**Strengths:**

- The paper demonstrates a commendable effort in addressing the problem at hand and provides a satisfactory solution.

**Weaknesses:**

- The problem being tackled seems incremental over the existing literature on distributed inverse constrained reinforcement learning.

**Questions:**

1. In reference to line # 43-44: Is it true that there is no existing work on online IRL with non-linear rewards? On a quick search here [1] is a work that works with quadratic rewards in online IRL.

2.  For better readability, I would suggest replacing 'state-of-the-arts' with 'state-of-the-art works' throughout the paper.

3. In reference to line # 66-67: What are the exact challenges in extending single-loop bi-level centralized optimization to de-centralized case?

4. What is `N' in line #76?

5. Is the paper the first one to propose a  'distributed online bi-level optimization problem'?

6. What are downstream applications to the problem setup discussed in para 3 of Introduction? Is the main challenge in ICRL estimating local rewards with streaming data. What are the exact challenges in this?

7. In general in dual optimization implementations, inner or outer optimization problems are not solved fully, how important is it to MA-BIRDS to solve any or all of the optimization problems fully?

[1] Self, R., Abudia, M., & Kamalapurkar, R. (2020, July). Online inverse reinforcement learning for systems with disturbances. In 2020 American control conference (ACC) (pp. 1118-1123). IEEE.

**Limitations:**

The authors include just one limitation about cost functions being linear. It would be nice to add a more comprehensive literature review.

---

> ### Author Rebuttal · Authors · 2023-08-02
>
> Thank you for your constructive feedback. We believe that this discussion will help improve our paper. We address your comments below:
>
> **Weakness: The problem being tackled seems incremental over the existing literature on distributed inverse constrained reinforcement learning (D-ICRL).**
>
> **Answer**: As discussed in the introduction, this paper extends D-ICRL to online learning, so the problem is incremental. However, the algorithm and analysis are substantially different from D-ICRL. Mentioned in lines 41-42, we include a section in Appendix to discuss our improvements from D-ICRL. In Appendix (section 7), we point out that our algorithm improves D-ICRL in almost every aspect, including weaker assumption, simpler and more efficient algorithm, much stronger theoretical and empirical results. Due to space limit, we cannot elaborate it here and please refer to Appendix (section 7) for details.
>
> **Q1: In reference to line 43-44: Is it true that there is no existing work on online IRL with non-linear rewards? On a quick search there is a work that works with quadratic rewards in online IRL.**
>
> **Answer**: We appreciate that you provide this reference and we will revise the statement in lines 43-44. We would like to add this reference and include a more comprehensive literature review.
>
> **Q2: For better readability, I would suggest replacing 'state-of-the-arts' with 'state-of-the-art works' throughout the paper.**
>
> **Answer**: Thanks for your advice, we will use "state-of-the-art works" instead.
>
> **Q3: In reference to line 66-67: What are the exact challenges in extending single-loop bi-level centralized optimization to de-centralized case?**
>
> **Answer**: It is very insightful that you point this out. The challenge mainly comes from the fact that the distributed learners do not know the global demonstration set but need to solve a global problem (3)-(4). For example, a key step of solving bi-level optimization problems is to compute the hyper-gradient (5). The hyper-gradient (5) contains an inverse-of-Hessian term $[\nabla_{\omega\omega}^2G]^{-1}$ (see the expression of $M(\theta,\omega)$ in line 205) that requires global demonstrations to compute because the function $G$ is global. A centralized learner can directly compute $[\nabla_{\omega\omega}^2G]^{-1}$ since it knows the global data. However, it is difficult for distributed learners to compute this term since $[\nabla_{\omega\omega}^2G]^{-1}$ is extremely hard to decompose.
>
> **Q4: What is `N' in line 76?**
>
> **Answer**: $N$ is the number of online iterations. We use $n$ to represent the general online iteration index and $N$ to represent a specific number of total online iterations.
>
> **Q5: Is the paper the first one to propose a 'distributed online bi-level optimization problem'?**
>
> **Answer**: To the best of our knowledge, there is no previous work on distributed online bi-level optimization.
>
> **Q6: What are downstream applications to the problem setup discussed in para 3 of Introduction? Is the main challenge in ICRL estimating local rewards with streaming data. What are the exact challenges in this?**
>
> **Answer**: Our problem setup is to learn reward functions and constraints from distributed and streaming demonstrations. There are plenty of downstream applications including inferring pedestrians' behaviors from their ongoing daily routines. The pedestrians have their destinations (i.e., reward functions) and some implicit constraints such as avoiding collision with obstacles. The demonstration data could be distributed since there could be multiple cameras recording the pedestrians' behaviors and the demonstrations are streaming given that the behaviors are recorded every day. Other applications include autonomous vehicles learning driving styles of humans from everyday observations and defenders learning the attacker's model from regular attacks in security, etc. The main challenge of our problem is to design fast and efficient algorithms that can learn reward functions and constraints from streaming data. Otherwise, the computation may not be finished before the next data arrives. Current works, e.g., double-loop methods or follow-the-leader methods, fail to update quickly at each online iteration. Therefore, we propose an online gradient descent single-loop algorithm (lines 53-55 and 64-65) that is much faster than baselines (see Tables 2-3).
>
> **Q7: In general in dual optimization implementations, inner or outer optimization problems are not solved fully, how important is it to MA-BIRDS to solve any or all of the optimization problems fully?**
>
> **Answer**: MA-BIRDS does not need to fully solve the outer or inner problem at any online iteration. In fact, this is the major contribution of MA-BIRDS. Fully solving the inner or outer problem will lead to more computation time, which is not suitable to be an online learning algorithm if the data arrives in a fast speed. MA-BIRDS only partially solves the inner and outer problems at each online iteration but we can still reach theoretical guarantees. More importantly, in the experiment, the baseline FTL fully solves the outer problem and the baseline DLM fully solves the inner problem. The empirical results show that MA-BIRDS is much faster than these two baselines in each online iteration and reaches the same good performance in a much shorter time (Tables 2-3).
>
> **Limitation: The authors include just one limitation about cost functions being linear. It would be nice to add a more comprehensive literature review.**
>
> **Answer**: We appreciate that you suggest a more comprehensive literature review. We will add more discussions on the related works, including online IRL, online non-convex optimization, and bi-level optimization. In specific, we will add the non-linear online IRL works and discuss our distinctions from them. For non-convex optimization and bi-level optimization, we will clearly mention that our work is the first one to study distributed online bi-level optimization.

---

### Official Review · Reviewer_UrMu · 2023-07-09

**Soundness:** 3 good
**Presentation:** 2 fair
**Contribution:** 3 good
**Rating:** 7
**Confidence:** 4

**Summary:**


The paper tries to infer the behaviors of multiple interacting experts by estimating their reward functions and constraints. In their approach, the distributed demonstrated trajectories are sequentially revealed to a group of learners. The problem is set up as a distributed online optimization problem with 2 levels - the outer-level estimates the reward functions  - the inner-level problem learns the constraints and corresponding policies. The solution is called MA-BIRDS - multi-agent behavior inference from distributed and streaming demonstrations algorithm. The results guarantee that 1) the distributed learners achieve consensus on reward functions, constraints, and policies; 2) the average local regret (over N online iterations) decreases, and the cumulative constraint violation increases sub-linearly. A couple of examples illustrate that the approach appears to work well.

**Strengths:**

The problem is very interesting - the approach seems sound, and the results (e.g. the 1) are solid with extensions of the SOA (e.g. [20-21]).

The simulation results compare to a good set of baselines

**Weaknesses:**

The learned results (constraints) in Figure 2 seems similar, but the visitation frequencies (colors) seems quite different - pls explain. Is the goal being learned as a constraint in fig 2(b) correct?

The results (fig 3) showed the perf compared to several centralized algorithms, but it would be good to provide evidence that the distributed learning problem is actually hard - ie that a naive distributed learner would fail in this case - would lead to strong results.

For the drone example, with only 2 goals and a demonstration presumably showing the path to the appropriate goal, it is difficult to see why this problem is hard? Pls provide more evidence that this is a difficult problem that you are succeeding on.

**Questions:**

Are the experts of similar capability? The more interesting case seems to be when they have varied capability and deciding amongst which to use of them?

Can the authors comments on the tightness of the bounds in the contribution statement - especially in the numerical results section?

**Limitations:**

some discussion

---

> ### Author Rebuttal · Authors · 2023-08-08
>
> Thanks for your constructive review. We appreciate that you find this work interesting and sound, and we believe that this discussion will contribute to a better paper. We address your comments below:
>
> **Weakness 1: The learned results (constraints) in Figure 2 seems similar, but the visitation frequencies (colors) seems quite different - pls explain. Is the goal being learned as a constraint in fig 2(b) correct?**
>
> **Answer**: The major reason of these different visitation frequencies is that the experts and learners follow different kinds of policies. The experts are programmed to follow the optimal policy (mentioned in line 290) while the learners use constrained soft Bellman policy (mentioned in Lemma 1). The expression of the constrained soft Bellman policy is included in Appendix (Subsection 8.1). From the expression, we can see that the constrained soft Bellman policy is more stochastic, e.g., it has non-zero probability of choosing any action and the probability of choosing two actions which have similar $Q$ values is similar. However, the optimal policy will only choose actions that have the highest $Q$ value. Therefore, it is expected that the constrained soft Bellman policy will lead to a more uniformly distributed visitation frequency. This aligns with the results in Figure 2. In Figure 2, the yellow states are must-visit states because they occupy the only exit of each corner. For those states that are not must-visit states, we can see that they are all green in Figure 2(b), i.e., the visitation frequency is more uniformly distributed, because the learners use the constrained soft Bellman policy. In practice, we use $\epsilon$-greedy policy for the experts and that's why some states that are not likely visited also have non-zero visitation frequency in Figure 2(a).
> The goal is to learn both the reward function and constraints so that the learners can behave like the experts. We use the visitation frequency to show that the learners can reach the destinations and avoid the obstacles just like the experts.
>
> **Weakness 2: The results (fig 3) showed the perf compared to several centralized algorithms, but it would be good to provide evidence that the distributed learning problem is actually hard - ie that a naive distributed learner would fail in this case - would lead to strong results.**
>
> **Answer**: Thank you for pointing this out. We include the comparison result between MA-BIRDS learners and naive distributed learners of the evader-patroller experiment in the newly uploaded PDF file (Figure 1). The naive learners only know local data and do not cooperate with each other. From the figure, we can see that the learners of MA-BIRDS perform much better than the naive distributed learners. The reason is that the naive distributed learners do not leverage communications to obtain information in the data received by the other learner while the distributed learners of MA-BIRDS can utilize communications to gain the information of the global data.
>
> **Weakness 3: For the drone example, with only 2 goals and a demonstration presumably showing the path to the appropriate goal, it is difficult to see why this problem is hard? Pls provide more evidence that this is a difficult problem that you are succeeding on.**
>
> **Answer**: We agree that given a demonstrated trajectory that shows the path to the goal, it is easy for humans to know the goal position. However, the learners aim to learn the reward function which includes information more than the goal position. For example, the reward function can capture the quantitative preference of the experts such as minimum time to reach the goal. This leads to the learned reward function giving significant large reward at the goal because the sooner the learners reach the goal, the longer they can stay at the goal and thus the larger reward they get (given that the time horizon $T$ is fixed). The learned reward function also captures qualitative preference such as do not stay too close to the obstacles and do not incline to go through the area between the obstacles. These result in the learned reward function giving lower reward in the area between the obstacles and the area around the obstacles. We can see that the learned trajectories in Figure 4(b) use short paths to the goal, stay a certain distance from the obstacles, and do not go through the area between the obstacles.
>
> **Q1: Are the experts of similar capability? The more interesting case seems to be when they have varied capability and deciding amongst which to use of them?**
>
> **Answer**: Yes, you are correct. The experts have similar capability. The situation you mention is interesting and we will investigate on this in future works.
>
> **Q2: Can the authors comments on the tightness of the bounds in the contribution statement - especially in the numerical results section?**
>
> **Answer**: Mentioned in lines 260-267, the bound has similar properties with the tight local regret bound proved in the literature [A1,A2]. We will add this discussion in the contribution statement. For the numerical section, instead of choosing a fixed time window length, the time window length $l$ increases with the number of online iterations (i.e., $l(n)=n$) (line 278). Therefore, the bound of the local regret diminishes to zero (mentioned in lines 263-264). Then the algorithm becomes a no-regret algorithm and the numerical results show that the learners can achieve similar performance to the experts when the number of online iterations increases.
>
> [A1] E. Hazan, K. Singh, and C. Zhang, "Efficient regret minimization in non-convex games,” International Conference on Machine Learning, pp. 1433–1441, 2017.
>
> [A2] N. Hallak, P. Mertikopoulos, and V. Cevher, “Regret minimization in stochastic non-convex learning via a proximal-gradient approach,” International Conference on Machine Learning, pp. 4008–4017, 2021.

---

### Author Rebuttal · Authors · 2023-08-08

Dear Reviewers,

Thanks for your detailed and constructive reviews. We believe that these discussions will lead to a stronger paper in general. We address all the comments of each reviewer under each review. We also include a PDF file in the global rebuttal that contains additional experiment results.

Best regards,
Authors.

---

### Decision · Program_Chairs · 2023-09-21

**Decision:**

Accept (poster)

**Comment:**

The paper studies the problem of inferring the behaviors of multiple interacting experts when demonstrations are streamed online. After rebuttal, there is general consensus among the reviewers about the interest of the setting and that the proposed approach is technical sound and it achieves satisfactory empirical performance. For all these reasons, I propose acceptance for the paper.

I would still like to encourage the authors to properly integrate their rebuttal in the final version of the paper. The motivation and the technical difficulties compared to previous work in a similar setting were much better clarified in the rebuttal than in the original submission. Similarly, it is important the authors include more intuition about the theoretical results and how to make the algorithm efficient.